# Efficacy and Safety of COVID-19 Vaccine in Patients on Renal Replacement Therapy

**DOI:** 10.3390/vaccines10091395

**Published:** 2022-08-25

**Authors:** Michela Frittoli, Matthias Cassia, Alessandra Barassi, Paola Ciceri, Andrea Galassi, Ferruccio Conte, Mario Gennaro Cozzolino

**Affiliations:** 1Renal Division, Department of Health Sciences, ASST Santi Paolo e Carlo, University of Milan, 20142 Milan, Italy; 2Laboratory of Clinical Biochemistry, Department of Health Sciences, ASST Santi Paolo e Carlo, University of Milan, 20122 Milan, Italy

**Keywords:** COVID-19, vaccine, humoral response, CKD, RRT, safety, dialysis, kidney transplantation

## Abstract

Patients with CKD on RRT are at high risk for severe disease and mortality in COVID-19 disease. We decided to conduct an observational prospective study to evaluate antibody response after vaccination for COVID-19 in a cohort of 210 adult patients on RRT (148 on HD; 20 on PD; and 42 kidney transplant recipients). Blood samples were taken before and 4 weeks after vaccination. Antibody levels were evaluated with CLIA immunoassay testing for IgG anti-trimeric spike protein of SARS-CoV-2. A positive antibody titer was present in 89.9% of HD patients, 90% of PD patients, and 52.4% of kidney transplant recipients. Non-responders were more frequent among patients on immunosuppressive therapy. Mycophenolate use in kidney transplant patients was associated with lower antibody response. The median antibody titer was 626 (228–1480) BAU/mL; higher in younger patients and those previously exposed to the virus and lower in HD patients with neoplasms and/or on immunosuppressive therapy. Only two patients developed COVID-19 in the observation period: they both had mild disease and antibody titers lower than 1000 BAU/mL. Our data show a valid response to COVID-19 mRNA vaccination in HD and PD patients and a reduced response in kidney transplant recipients. Mycophenolate was the most relevant factor associated with low response.

## 1. Introduction

Patients with chronic kidney disease (CKD) and on renal replacement therapy (RRT) are not only at particularly increased risk for severe Coronavirus disease 2019 (COVID-19), but also have particular susceptibility to COVID-19 infection [1]. These patients are usually highly medicalized, frequently undergoing in-hospital RRT thrice weekly or at least periodical medical monitoring, and have multifactorial immune deficiency secondary to CKD [2,3,4]. Moreover, CKD has been identified as a major risk factor for severe disease and mortality in patients with COVID-19 [5,6]. Considering these points, since the development of effective vaccines, efforts have been made to vaccinate this particularly vulnerable population with priority. However, it is known that patients with CKD have a lower rate of response to vaccines and they are usually excluded from clinical trials: some doubts about their response have been raised since SARS-CoV-2 vaccines’ initial diffusion [7,8,9]. Given these data, we conducted an observational prospective study on a cohort of CKD patients on renal replacement therapy to evaluate antibody response after a complete mRNA vaccination against COVID-19 and its correlation with clinical protection from infection.

## 2. Materials and Methods

### 2.1. Population of the Study

All adult patients on chronic renal replacement therapy, including patients receiving maintenance hemodialysis, peritoneal dialysis, and kidney transplant recipients, who completed a cycle of mRNA anti-COVID-19 vaccination in our center (ASST Santi Paolo e Carlo, Milan, Italy) were included in the study. Exclusion criteria were clinical evidence of COVID-19 infection in the prior 3 months, hospitalization for any reason between the two doses, and the delay of the second dose administration for any reason. For all patients, clinical data from medical records were collected with particular reference to comorbidities, etiology of kidney disease, renal replacement modality, and dialysis vintage. For kidney transplant recipients, creatinine clearance on 24 h urine samples was also measured to evaluate the stage of chronic kidney disease if present. For all patients, any prior COVID-19 infection, defined as a positive PCR test performed for any reason before the first vaccine dose, and any immunosuppressive therapy were also recorded.

### 2.2. Methods

All patients were vaccinated between March and May 2021, with two doses of mRNA anti-COVID-19 vaccine, following the recommendations on the data sheets. According to regional allocation policies, patients on hemodialysis and peritoneal dialysis received BNT162b2 in two 0.3 mL doses, 21 days apart, while kidney transplant recipients received mRNA-1273 in two 0.5 mL doses, 28 days apart. A serum sample was drawn from each patient on 2 different occasions: the first sample was taken just before the administration of the first vaccine dose, the second one 4 weeks after the second dose. Blood samples were centrifuged at 3500 rpm for 10 min and the resulting plasma was frozen at −20 °C until the completion of the sampling, when they were analyzed for specific IgG quantification. The serological test was performed using LIAISON SARS-CoV-2 TrimericS IgG assay (DiaSorin, Saluggia, Italy), a chemiluminescence immunoassay for the quantitative determination of anti-trimeric spike protein-specific IgG antibodies to SARS-CoV-2 in human serum or plasma samples. The trimeric spike glycoprotein is a stabilized trimer whose antibodies showed a high correlation with the microneutralization test in vitro [10,11]. The assay is fully automated; sensitivity and specificity are 98.7% and 99.5%, respectively. Results are expressed in BAU/mL, with a range from 4.81 BAU/mL to 2080 BAU/mL with higher levels requiring automatic dilution at 1:20. The positivity cut-off is set at 33.8 BAU/mL. Patients were also clinically observed for six months after completing the anti-COVID-19 vaccine cycle, with particular reference to SARS-CoV-2 infection and/or symptoms referable to side effects of the vaccine.

### 2.3. Statistical Analysis

Quantitative variables were described as mean ± standard deviation or median (interquartile range) according to their distribution (normal or non-normal, respectively). Student’s *t*-test and Wilcoxon–Mann–Whitney test were used for comparison of normally distributed and non-parametric variables, respectively. Qualitative variables were described as frequency (%) and analyzed using the Chi-squared test and Fischer’s exact test. Correlations between quantitative variables were evaluated using linear regression. All statistical analyses were performed using IBM SPSS Statistics software (version 28.0.0.0). The level of statistical significance was set at <0.05.

## 3. Results

### 3.1. Population Characteristics

Based on inclusion and exclusion criteria, 210 patients were included in our study during the study period from March to May 2021. Among them, 148 were on hemodialysis, 20 were on peritoneal dialysis, and 42 were kidney transplant recipients. The mean age in the hemodialysis group was 69 years, while it was lower in both the peritoneal dialysis group (63 years) and the transplant group (60 years). Sex distribution was similar among all groups, with 65.5% of males. The most common comorbidities (Table 1) were hypertension, diabetes, and heart disease. All dialysis patients had a negligible residual renal function, while transplant patients had a mean creatinine clearance of 54.9 ± 20.5 mL/min. While immunosuppressive therapy was used in only 5.9% of dialysis patients, all transplant recipients were on immunosuppressive therapy. Among the transplanted patients, the most commonly used immunosuppressive drug was steroid (80.9%), followed by cyclosporine (71.4%) and mycophenolate (71.4%). The majority of the patients used three different immunosuppressive drugs, while 25% of them were on a double regimen or a single drug. Prior COVID-19 infection was reported in 6 patients total: 3 in the hemodialysis group and 3 in the transplant group.

### 3.2. Antibody Response

A positive antibody titer was present at the end of the vaccination in 89.9% of hemodialysis patients, 90% of peritoneal dialysis patients, and 52.4% of kidney transplant recipients (*p* < 0.001) (Table 2). Non-responders of all patients were more frequent among those on immunosuppressive therapy (*p* < 0.001) and with hypertension (*p* = 0.022), while all patients with a previous antibody titer maintained it after vaccination. It is interesting to note that only 6 patients had a clinically relevant previous COVID-19 infection, but 20 patients had a positive pre-vaccination titer, probably accounting for previous infections with a sub-clinical course. Moreover, among hemodialysis patients, non-responders were significantly older (*p* = 0.003) and more often diabetics (*p* = 0.020) and on immunosuppressive treatment (*p* = 0.017).

For kidney transplant recipients, no difference was found in the distribution of responders and non-responders between patients on three immunosuppressive medications and the ones on two or one drug. However, patients on mycophenolate were more frequently non-responders (*p* = 0.011) (Table 3).

### 3.3. Antibody Titer

The median antibody titer among responders was 626 (228–1480) BAU/mL. Antibody titer was found to be higher in younger patients (*p* = 0.032) and patients previously exposed to the virus and/or with a positive titer before vaccination (*p* = 0.034 and *p* < 0.001). Furthermore, antibody levels were significantly lower in patients on hemodialysis affected by malignancies (*p* = 0.021) and on immunosuppressive therapy (*p* = 0.014). Among kidney transplant responders, lower antibodies were found in patients on steroid therapy, although with a non-significant difference (*p* = 0.053). No other drug or anamnestic characteristic was correlated with higher or lower antibody response. Values (median and interquartile range) of post-vaccine antibody titer in responders are presented in Table 4.

### 3.4. Safety and Protection

Vaccination was overall well tolerated in all patients: none of the patients reported serious side effects, as confirmed by a recent meta-analysis [12]. Common side effects were self-limiting fever, pain at the injection site, and fatigue, none of which required treatment other than short-course analgesics.

During the six months observation, only two patients developed COVID-19 infection: the first one was a 65-year-old woman on hemodialysis treatment who tested positive for a PCR swab after developing fever and myalgias 3 months after vaccination; the second one was a 68-year-old man on peritoneal dialysis with similar symptoms who tested positive 6 months after vaccination. Both patients had a self-limiting disease requiring only supportive therapy; none of them required hospitalization or oxygen supplementation, however, the second one was treated with Casirivimab/Imdevimab as an outpatient. Both patients were among responders: the first one had a 167 BAU/mL titer, while the second one had an antibody response of 737 BAU/mL.

It was hypothesized that a peak titer of at least 1000 BAU/mL could be necessary for COVID-19 protection. Indeed, similarly to what was found for response predictivity, responders with antibody levels >999 BAU/mL were younger than the ones with lower titers (*p* = 0.003) and had more frequent prior exposure to the virus (*p* < 0.001), while titers <100 BAU/mL were more frequent in patients on immunosuppressive therapy (*p* = 0.005).

## 4. Discussion

Our data show a valid response to COVID-19 mRNA vaccination in hemodialysis and peritoneal dialysis patients (90%), and a significantly reduced response in kidney transplant recipients (52.4%). Data from dialysis patients are consistent with data from Yanay et al., Grupper et al., Agur et al., and Frantzen et al. [13,14,15,16]. However, there are also some studies showing lower response rates [17,18], particularly when the antibody evaluation was made earlier than 30 days from vaccination [19,20,21,22]. Despite this consistency, it is important to address the huge differences among different studies in terms of timing of antibody evaluation and immunoassays used: these limitations make data not comparable, and the standardization of the procedures used is needed.

Data on kidney transplant recipients are even more variable among the different studies and, with the same limitations as discussed before, they show lower response rates than what we observed, varying from 25% to 48%, with some studies reporting response rates as low as 3% [18,23,24,25,26]. This difference could be explained by the fact that there were nine patients in our transplant cohort with a detectable antibody titer before vaccination, a strong predictor of response. Seroconversion in this group was 39.4%, similar to what was observed in other studies [27]. Low antibody response in these patients, thus exposing them due to their specific frailty to an increased risk of developing COVID-19 infection, should suggest the need for modification of the vaccine strategy with more suitable schedules [28].

In terms of factors associated with lack of response, immunosuppressive therapy was the most relevant, in particular for mycophenolate (Table 3). Mycophenolic acid is a reversible inhibitor of the inosine monophosphate dehydrogenase (IMPDH), an enzyme essential to the de novo synthesis of guanosine-triphosphate (GTP) in lymphocytes, and like azathioprine presents an anti-proliferative effect on lymphocytes. While some studies show lower rates of response in patients on belatacept and with higher calcineurin inhibitor levels [25,29,30], only a few studies reported similar data on mycophenolate [25,31,32]. Based on this result, a reduction or even temporary discontinuation of mycophenolate dose could be hypothesized when a booster dose of vaccine to previously non-responder patients is administered. This hypothesis wasformulated as a result of Connolly’s study [33] in patients with Rheumatoid Arthritis, which was later not confirmed in the nonrandomized pilot study of Florina Regele [34], with the discontinuation of mycophenolate therapy in the 2 weeks pre-vaccination in transplant patients.

Hypertension was the other factor associated with lack of response. This finding is also difficult to comment on because of the paucity of data in the literature. In our patient cohort, a reduced antibody titer was present in all hypertensive patients at univariate analysis (Table 2), but in both subgroups of HD and transplant patients, this difference did not reach statistical significance. Only in PD patients was there a significant association between hypertension and antibody response (*p* = 0.02). The data in the literature confirming this reduced response are discordant, with some studies [35,36] confirming the finding and others [31,37,38,39] highlighting the loss of significance of the finding when the variable is analyzed in a multivariate or age-adjusted analysis [38,39].

When evaluating protection from infection, data from our study are insufficient to draw any conclusion. We observed only two cases of COVID-19 after vaccination, both mild and self-limiting, and both in patients who were responders, with no characteristics associated with lower response. Both patients had antibody titers <1000 BAU/mL, suggesting that higher levels could be required for clinical protection. However, there were no cases of infection among non-responders and among patients with comorbidities and therapies associated with lack of response, which probably explains the importance of other protective measures such as self-isolation and personal and general protective systems used in our unit.

## 5. Conclusions

This study provides some evidence of a valid response to the COVID-19 vaccine in CKD patients on dialysis while pointing out a reduced response in kidney transplant recipients. However, our observational study has some limitations, such as the limited number of patients enrolled, the short observation period, being a single-center study, and the different use of BNT162b2 and mRNA-1273 vaccines between dialysis and transplant patients, according to vaccine allocation protocols. BNT162b2 and mRNA-1273 are reported as similar in terms of efficacy [40], but, interestingly, a study by Kaiser et al. showed superior immunogenicity of mRNA-1273 in dialysis patients [41,42]. Given this finding, the lower rate of response in our transplant group, vaccinated with mRNA-1273, can be even more concerning. On the other hand, the authors, while being aware of a large number of publications regarding the topic of this study, are also aware that this is one of the few studies in the field involving all modalities of RRT treated by a single nephrology department, thus ensuring homogeneity of management and potentially better reliability of results. Comparing with the studies reviewed in the recent meta-analysis by Ma B.M. et al. [12] of the 10 studies used to assess antibody response to COVID-19 vaccination, only one was representative of all RRT modalities with a lower population size.

Since immunosuppressive therapy, and mycophenolate, in particular, has been associated with a lack of response to the COVID-19 vaccine in transplant recipients, it could be hypothesized that dose reduction or temporary discontinuation of mycophenolate before a booster dose of the vaccine could improve the serologic response. Unfortunately, studies to date are discordant, probably also related to the too short duration of the proposed discontinuation for the effect of mycophenolate to cease inhibition of the immune response. However, the data from our study are insufficient to recommend changes in immunosuppressive therapies based on the serologic response, and further studies will be needed to give an unambiguous level of antibody titer efficacy in preventing or attenuating severe SARS-CoV disease. Finally, we observed that both dialysis patients who developed COVID-19 after vaccination had positive antibody levels, although below 1000 BAU/mL. We suggested that higher antibody titers may be needed for protection in this particular population, but further studies, including mid-to-long-term observation, will be needed to define a sufficient antibody level for clinical protection. This endpoint should be a priority because it could guide preventive measures and resource allocation in dialysis units and hospitals.

## Figures and Tables

**Table 1 vaccines-10-01395-t001:** Population characteristics.

Patients Clinical Features		HD	PD	TX
Age (years)		69 ± 12	63 ± 11	60 ± 11
Male% (*n*)		66% (98)	60% (12)	69% (29)
Treatment vintage (months)		59.3 ± 55.3	37 ± 17	12.6 ± 7.9
ESRD etiology % (*n*)	Not Known	26.4% (39)	15% (3)	23.8% (10)
	Diabetes	25.7% (38)	15% (3)	2.4% (1)
	Hypertensive	17.6% (26)	20% (4)	14.3% (6)
	Urologic disease	6.8% (10)	-	9.5% (4)
	ADPKD	5.4% (8)	20% (4)	2.4% (1)
	IgA N	3.4% (5)	15% (3)	16.7% (7)
	Membranous N.	2.7% (4)	-	-
	MPGN	2% (3)	-	7.1% (3)
	FSGS	-	-	11.9% (5)
	Myeloma	2% (3)	-	-
	Amyloidosis	1.4% (2)	-	-
	Vasculitis	1.4% (2)	-	-
	Other genetic diseases	-	-	7.1% (3)
	Others	5.4% (8)	15% (3)	4.8% (2)
Comorbidities % (*n*)	Hypertension	86.5% (128)	95% (19)	97.6% (41)
	Diabetes	33.1% (49)	20% (4)	19% (8)
	Heart disease	31.8% (47)	30% (6)	7.1% (3)
	COPD	20.9% (31)	-	-
	Neoplasia	11.5% (17)	-	-
Immunosuppressive therapy % (*n*)	Steroids			80.9% (34)
	Cyclosporine			71.4% (10)
	Tacrolimus			23.8% (10)
	Mycophenolate			71.4% (30)
	Azathioprine			11.9% (5)
	Everolimus			11.9% (5)
	Unspecified	6.1% (9)	5% (1)	-

ADPKD: Autosomal Dominant Polycystic Kidney Disease; MPGN: Membranoproliferative Glomerulonephritis; FSGS: Focal segmental glomerulosclerosis; COPD: Chronic obstructive pulmonary disease.

**Table 2 vaccines-10-01395-t002:** Vaccine responsiveness and population characteristics.

Patients Clinical Features			Non-Responders	Responders	*p*
**Age (years)**			68 ± 13.9	66.4 ± 12.7	ns
**Gender**		Males	18% (25)	82% (114)	ns
		Females	16.9% (12)	83.1% (59)	
**RRT**		HD	10.1% (15)	89.9% (133)	<0.001
		PD	10% (2)	90% (18)	
		TX	47.6% (20)	52.4% (22)	
**Comorbidities**	Hypertension % (*n*)	Yes	19.7% (37)	80.3% (151)	0.022
		No	0	100 (22)	
	Heart disease%(*n*)	Yes	12.5% (7)	87.5% (49)	ns
		No	19.5% (30)	80.5% (124)	
	Immunosuppressive therapy% (*n*)	Yes	44.2% (23)	55.8% (29)	<0.001
		No	8.9% (14)	91.1% (144)	
	Prior COVID-19	Yes	0	100% (6)	ns
		No	18.1% (37)	81.9% (167)	
	Pre-vaccine titer%(*n*)	Positive	0	100% (20)	0.030
		Negative	19.5% (37)	80.5% (153)	

ns = not significant.

**Table 3 vaccines-10-01395-t003:** Immunosuppressive therapy and vaccine responsiveness.

		Non-Responders	Responders	*p*
Immunosuppressive therapy	3 drugs	51.6% (16)	48.4% (15)	ns
	2 drugs	36.4% (4)	63.6% (7)	
Steroids	Yes	47.1% (16)	52.9% (18)	ns
	No	50% (4)	50% (4)	
Cyclosporine	Yes	50% (15)	50% (15)	ns
	No	41.7% (5)	58.3% (7)	
Tacrolimus	Yes	50% (5)	50% (5)	ns
	No	46.9% (15)	53.1% (17)	
Mycophenolate	Yes	60% (18)	40% (12)	0.011
	No	16.7% (2)	83% (18)	
Azathioprine	Yes	20% (1)	80% (4)	ns
	No	51.4% (19)	48.6% (18)	
Everolimus	Yes	20% (1)	80% (4)	ns
	No	51.4% (19)	48.6% (18)	

ns = not significant.

**Table 4 vaccines-10-01395-t004:** Antibody titers in responders expressed as median (interquartile range).

Patients Clinical Features			Ab Titer (BAU/mL)	*p*
Age (years)				0.016
Gender		Males	600 (1582.5–223)	0.958
		Females	663 (1400–229)	
RRT		HD	537 (39,600–36.3)	0.055
		PD	775 (9610–103)	
		TX	1050 (36,300–35.8)	
Comorbidities	Hypertension	Yes	616 (1440–229)	0.748
		No	910 (1720–169.75)	
	Heart disease	Yes	600 (1490–285)	0.745
		No	644 (1592.5–221)	
	Diabetes	Yes	559 (1190–210)	0.519
		No	645 (1592.5–221)	
	Immunosuppressive therapy	Yes	821 (6605–94.75)	0.535
		No	605 (1437.5–229.5)	
	Prior COVID-19	Yes	1550 (11,047.5–729.75)	0.034
		No	600 (1450–224)	
	Pre-vaccine titer	Positive	7885 (11,485–1967.5)	<0.001
		Negative	485 (1190–209.5)	

## Data Availability

The data presented in this study are available, if needed, on reasonable request in anonymous form from the authors.

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
