# Peer review of "Efficacy and Safety of COVID-19 Vaccine in Patients on Renal Replacement Therapy"

_vaccines, 2022, doi:10.3390/vaccines10091395_

Round 1

Reviewer 1 Report

This article presents a single centres serology results to 2 doses of mRNA covid-19 vaccination in patients receiving dialysis or with a kidney transplant. The main issues with the paper is a lack of novelty as there are already many papers presenting this information available.

Other comments

1. There is a difference in vaccine dose and dosing interval between dialysis and transplant patients which precludes direct comparison. Why were the different doses and regimens selected for use?

2. Against which omicron variants were the antibodies titred. Were any VOC tested? It appears neutralisation assays were not used? Were anti-RBD titres considered/measured?

3. There are 6 patients said to have covid prior to vaccination but 20 with positive titres. If they hadn't been vaccinated prior to this study how is it proposed that they developed antibodies? Perhaps there were cases of subclinical infection? Would be good to have this more clearly defined.

4. Was the effect of GFR assessed in transplant patient seroresponses?

Author Response

This article presents a single centres serology results to 2 doses of mRNA covid-19 vaccination in patients receiving dialysis or with a kidney transplant. The main issues with the paper is a lack of novelty as there are already many papers presenting this information available.

Other comments

  1. There is a difference in vaccine dose and dosing interval between dialysis and transplant patients which precludes direct comparison. Why were the different doses and regimens selected for use?

The authors thank the reviewer for the questions that provide an opportunity for a more thorough explanation.

  1. Dialysis and transplant patients were vaccinated with two different vaccines according to the organizational procedures of the Lombardy Region Health Department. Dialysis patients have been immunized in dialysis units where BNT162b2 was available according to regional allocation. Transplant patients were vaccinated as part of an outpatients vaccination program to which mRNA-1273 was allocated. Different doses and intervals were defined by the data sheets of the two different vaccines. BNT162b2 and  mRNA-1273, however, are both mRNA vaccines with reported similar clinical efficacy (1). There are, however, some data regarding the superior immunogenicity of mRNA-1273 in dialysis patientsThe immunogenicity of the mRNA-1273 vaccine is markedly better than that of the BNT162b2 vaccine, most likely by virtue of its higher mRNA content” (2,3). Lower rates of response in transplant patients vaccinated with mRNA-1273 could be considered even more concerning given these data. The authors are aware of the difference in the 2 types of vaccines used for dialysis and transplant patients, but as stated in the paper, this study is observational and not interventional, and as such should follow the recommendations prescribed by health authorities
  2. Against which omicron variants were the antibodies titred. Were any VOC tested? It appears neutralisation assays were not used? Were anti-RBD titres considered/measured?
  3. At the time of the survey (March-May 2021), knowledge about circulating variants was scarce, at least in our country, and no tests were available. We did not use neutralization tests since the main purpose of the study is to provide clinical data. However, we chose a serological test for anti-trimeric spike protein and not anti-RBD because this specific test used showed a high correlation with the in vitro microneutralization test, as reported in "Methods."
  4. There are 6 patients said to have covid prior to vaccination but 20 with positive titres. If they hadn't been vaccinated prior to this study how is it proposed that they developed antibodies? Perhaps there were cases of subclinical infection? Would be good to have this more clearly defined.
  5. It is highly probable that these patients had an infection with a sub-clinical course, since patients were not routinely tested if asymptomatic. We added a specification in the text.
  6. Was the effect of GFR assessed in transplant patient sero responses?
  7. Yes, we did, but it was not significant. In the transplant group GFR was not different between responder and non-responders (p=0,110) and among responders there was no correlation between GFR and antibody titres (p=0,498).
  8. About Novelty:

    Reviewer 1 drew attention to the lack of novelty of the study submitted to you entitled "Efficacy and Safety of Covid-19 Vaccine in Patients on Renal Replacement Therapy.” While the authors are aware of a large number of studies concerning the subject of the study, they consider it useful for the acceptance for publication to highlight some peculiarities in comparison with other published work.

    1. It must be kept in mind that this work is, as stated, "observational" and therefore representative of the real world in which health care professionals operate towards patients with ESRD.
    2. The study simultaneously highlights the immune response to Covid-19 miRNA vaccines of RRT patients in all treatment modalities from a single center, with homogeneity of management. From a survey of the most recent literature, the meta-analysis by Ma BM et al.[1] evaluating the immune response emerging from several studies considers for the final analysis only 10 papers that are explicitly highlighted in the results: of these only one is representative of all treatment modalities HD, PD, and Transplantation, all other describing only partially. In addition, the paper submitted has a large sample size both overall and in individual modalities (the highest compared to those reported in the article cited above), which reinforces the significance of the results obtained.          
    3. Ma BM, Tam AR, Chan KW, Ma MKM,Hung IFN, Yap DYH and Chan TM (2022) Immunogenicity and Safety ofCOVID-19 Vaccines in Patients Receiving Renal Replacement Therapy: A Systematic Review and Meta-Analysis. Front. Med. 9:827859.doi: 10.3389/fmed.2022.827859

Reviewer 2 Report

The authors touch hot topic of sensitive population with CKD and their response to the vaccination. I mostly missed the novelty, found some drawbacks in methodology: how could you explain the exclusion criteria: hospitalization between two doses for any reason. This gives we some concern. What is etiology of cKD named as genetic disease as some other mentioned like ADPKD are genetic as well. 

No explanation of abbreviations : GSFS, BPCO. 

The statement, that patients with hypertension had weaker antibody response seem not motivated. 

I missed raw data of antibodies of the patients, the ranges.  

Author Response

The authors touch hot topic of sensitive population with CKD and their response to the vaccination. I mostly missed the novelty, found some drawbacks in methodology: how could you explain the exclusion criteria: hospitalization between two doses for any reason. This gives we some concern. What is etiology of cKD named as genetic disease as some other mentioned like ADPKD are genetic as well.

The authors thank the reviewer for the questions that do the chance for a better explanation

  1. Patients hospitalized between the two doses were excluded because diseases (mostly infections) and therapies could have been biased for the immunological response.
  2. ADPKD is the most common genetic form of kidney disease so we made a separate group for it; “genetic disease” in the table stands for other genetic diseases grouped (it includes Alport disease, Tuberous sclerosis, and tubular disorders). We changed the term in “Other genetic diseases”

No explanation of abbreviations : GSFS, BPCO.

  1. Explanations were added, for the typing error (focal segmental glomerulosclerosis =FSGF, Chronic obstructive pulmonary disease =COPD)

The statement, that patients with hypertension had weaker antibody response seem not motivated.

  1. Patients with hypertension had lower rates of response when analyzing the whole cohort, but this finding was confirmed only in the peritoneal dialysis sub-group, being non-significant in the HD and transplant sub-groups. We have no clear explanation for this finding so we presented it as it is, reporting the limited data available in the literature; more large studies would probably clarify the reliability of this result.

I missed raw data of antibodies of the patients, the ranges.

  1. A specific table can be added 

    Ab titer (BAU/ml)

    P

    Age (years)

    linear

    0.016

    Gender

    Males

    600(1582.5-223)

    0.958

    Females

    663(1400-229)

    RRT

    HD

    537(39600-36.3

    0.055

    PD

    775(9610-103)

    TX

    1050(36300-35.8)

    Comorbidities

    Hypertension

    Yes

    616(1440-229)

    0.748

    No

    910(1720-169.75)

    Heart disease

    Yes

    600(1490-285)

    0.745

    No

    644(1592.5-221)

    Diabetes

    Yes

    559(1190-210)

    0.519

    No

    645(1592.5-221)

    Immunosuppressive therapy

    Yes

    821(6605-94.75)

    0.535

    No

    605(1437.5-229.5)

    Prior Covid-19

    Yes

    1550(11047.5-729.75)

    0.034

    No

    600(1450-224)

    Pre-vaccine titer

    Positive

    7885(11485-1967.5)

    <0.001

    Negative

    485(1190-209.5)

    Table 4. Antibody titers in responders expressed as median (interquartile range)

  2. About Novelty:

    Reviewer 2 drew attention to the lack of novelty of the study submitted to you entitled "Efficacy and Safety of Covid-19 Vaccine in Patients on Renal Replacement Therapy.” While the authors are aware of a large number of studies concerning the subject of the study, they consider it useful for the acceptance for publication to highlight some peculiarities in comparison with other published work.

    1. It must be kept in mind that this work is, as stated, "observational" and therefore representative of the real world in which health care professionals operate towards patients with ESRD.

The study simultaneously highlights the immune response to Covid-19 miRNA vaccines of RRT patients in all treatment modalities from a single center, with homogeneity of management. From a survey of the most recent literature, the meta-analysis by Ma BM et al. evaluating the immune response emerging from several studies considers for the final analysis only 10 papers that are explicitly highlighted in the results: of these only one is representative of all treatment modalities HD, PD, and Transplantation, and all others describing only partially. In addition, the paper submitted has a large sample size both overall and in individual modalities (the highest compared to those reported in the article cited above), which reinforces the significance of the results obtained.          

Ma BM, Tam AR, Chan KW, Ma MKM, Hung IFN, Yap DYH, and Chan TM (2022) Immunogenicity and Safety ofCOVID-19 Vaccines in Patients Receiving Renal Replacement Therapy: A Systematic Review and Meta-Analysis. Front. Med. 9:827859.doi: 10.3389/fmed.2022.827859

Reviewer 3 Report

In this report, the authors conducted an observational prospective study on a cohort of CKD patients on renal replacement therapy to evaluate antibody response after a com- plete mRNA vaccination against COVID-19 and its correlation with clinical protection 52 from infection.

They concluded that their data had shown a valid response to COVID-19 mRNA vaccination in HD and PD patients and a reduced response in kidney 33 transplant recipients, and that mycophenolate was the most relevant factor associated with low response.

I think it is interesting, but there is a major problem.

Major problem

Patients on hemodialysis and peritoneal dialysis received BNT162b2 in two 0.3 ml doses, 21 days apart, but kidney transplant recipients received mRNA-1273 in two 0.5 ml doses, 28 days apart. It seems meaningless to compare the effect of two different vaccines.

How do the authors interpret that different vaccines have different effects?

I think it’s better to compare responders and non-responders in patients on hemodialysis and peritoneal dialysis. Comparison of responders and non-responders in kidney transplant recipients should be separate from patients on hemodialysis and peritoneal dialysis.

Author Response

In this report, the authors conducted an observational prospective study on a cohort of CKD patients on renal replacement therapy to evaluate antibody response after a complete mRNA vaccination against COVID-19 and its correlation with clinical protection from infection.

They concluded that their data had shown a valid response to COVID-19 mRNA vaccination in HD and PD patients and a reduced response in kidney transplant recipients, and that mycophenolate was the most relevant factor associated with low response.

I think it is interesting, but there is a major problem.

Major problem

Patients on hemodialysis and peritoneal dialysis received BNT162b2 in two 0.3 ml doses, 21 days apart, but kidney transplant recipients received mRNA-1273 in two 0.5 ml doses, 28 days apart. It seems meaningless to compare the effect of two different vaccines.

How do the authors interpret that different vaccines have different effects?

I think it’s better to compare responders and non-responders in patients on hemodialysis and peritoneal dialysis. Comparison of responders and non-responders in kidney transplant recipients should be separate from patients on hemodialysis and peritoneal dialysis.

The authors thank the reviewers for the questions that do the chance for a better explanation

  1. Vaccines were allocated according to regional recommendations so we were not able to give the same vaccine to all patients. Given the data available, however, BNT162b2 and mRNA-1273 are both mRNA vaccines coding for SARS-CoV-2 spike protein with similar efficacy [1]. Some data are showing superior immunogenicity of mRNA-1273 in dialysis patients [2,3], which makes the lower response rate of transplant patients even more interesting.
  2. The most interesting finding, however, is about the effect of immunosuppressive therapy and in particular of mycophenolate. This finding comes from a sub-analysis of the transplanted patients, all vaccinated with mRNA-1273. As reported in the paper of  Florina Regele et al. [4] mycophenolic acid is a reversible inhibitor of the inosine monophosphate dehydrogenase (IMPDH), an enzyme essential to the de novo synthesis of guanosine-triphosphate (GTP) in lymphocytes. Short-term discontinuation of antiproliferative immunosuppression has been proposed as a strategy to increase response to SARS-CoV-2 vaccines in immunosuppressed individuals.

References

  1. Hahn WO, Wiley Z. COVID-19 Vaccines. Infect Dis Clin North Am. 2022 Jun;36(2):481-494. doi: 10.1016/j.idc.2022.01.008. Epub 2022 Jan 31. PMID: 35636910; PMCID: PMC8802612

  1. Kaiser RA, Haller MC, Apfalter P, Kerschner H, Cejka D. Comparison of BNT162b2 (Pfizer-BioNtech) and mRNA-1273 (Moderna) SARS-CoV-2 mRNA vaccine immunogenicity in dialysis patients. Kidney Int. 2021 Sep;100(3):697-698. doi: 10.1016/j.kint.2021.07.004. Epub 2021 Jul 13. PMID: 34270945; PMCID: PMC8276570

  1. Khalil El Karoui and An S. De Vriese COVID-19 in dialysis: clinical impact, immune response, prevention, and treatment Kidney International (2022) 101, 883–894; https://doi.org/10.1016/kint.2022.01.022

Round 2

Reviewer 1 Report

Need to provide evidence of human research ethics approval or waiver of need for such

Author Response

As already written in the second cover letter, about the approval of
the ethics board, since the study was exclusively observational and
limited to routine clinic-therapeutic interventions, for which patients
signed informed consent at the beginning of treatment for ESKD, no
further approval was necessary. In any case, a specific consent was
signed for sampling for the anti-Covid serological assay. Consensus
statements for both vaccination and serological sampling were previously
sent to Editorial team. In any case, from the contents of the study in
no way can the identity of the patients be traced, safeguarding their
privacy. 

Reviewer 3 Report

I understand the authors’ point of view.

Thus, it is assumed that a publication is possible.

Author Response

Thank you for your review and providing feedback.